# Microscopic Polyangiitis Presenting with Splenic Infarction: A Case Report

**DOI:** 10.3390/medicina57020157

**Published:** 2021-02-09

**Authors:** Sang Wan Chung

**Affiliations:** Division of Rheumatology, Department of Internal Medicine, School of Medicine, Kyung Hee University, 26 Kyungheedae-ro, Dongdaemun-gu, Seoul 02447, Korea; wanyworld83@gmail.com; Tel.: +82-2-958-8200

**Keywords:** ANCA-associated vasculitis, splenic infarction, MPA

## Abstract

Microscopic polyangiitis (MPA) is an anti-neutrophil cytoplasmic antibody (ANCA)-associated vasculitis (AAV). The splenic involvement in AAV is known to be rare, and that in MPA has not been reported to date. A 74-year-old woman was admitted owing to left arm numbness and weakness. The patient was diagnosed as MPA with vasculitic neuropathy. Her abdominal computed tomography (CT) revealed splenic infarction incidentally. The splenic infarction had been resolved at follow-up CT after treatment. If splenic involvement of MPA was not considered, treatment may have been delayed in order to differentiate other diseases. Herein, I report the first case of splenic involvement of MPA.

## 1. Introduction

Microscopic polyangiitis (MPA) is a type of antineutrophil cytoplasmic antibody (ANCA)-associated vasculitis (AAV) that predominantly affects small-sized vessels. The most commonly affected organs are the kidney and lung. Necrotizing and crescentic glomerulonephritis and diffuse alveolar hemorrhage are the main clinical presentations. General signs such as fever, weight loss, peripheral neuropathy, cutaneous manifestations, and upper respiratory tract involvement may be present [1]. In AAV numerous atypical manifestations may also occur, among which, though rarely reported, is splenic involvement, particularly in patients with granulomatous polyangiitis (GPA) [2]. In MPA, splenic involvement has been reported very rarely. Herein, I report a case of splenic involvement of MPA.

## 2. Case Presentation

A 74-year-old female patient presented to the emergency department of Kyung Hee University Medical Center because of dizziness and generalized weakness that started 2 days prior. The patient had a history of old cerebral infarction and was taking oral medications for hypertension and dyslipidemia. Brain magnetic resonance imaging (MRI) revealed an acute focal infarction in the right basal ganglia. At that time, in the laboratory examinations, the erythrocyte sedimentation rate (ESR) was 74 mm/h (normal range, <20 mm/h) and C-reactive protein (CRP) level was 6.61 mg/dL (<0.5 mg/dL) Aspirin was prescribed to the patient. One month later, the patient was admitted to our neurology clinic because of weakness in her left arm and numbness in both hands. MRI revealed no diffusion-restricted lesion in the brain. Physical examination revealed no remarkable findings except for paresthesia in the left hand. The patient’s motor power of four limbs were normal (grade 5). Her initial chest radiograph was normal. Her laboratory examination results showed a white blood cell count of 9.79 × 109/L (76% neutrophils), ESR of 120 mm/h, and CRP level of 18.44 mg/dL. However, she had no fever. Her serum creatinine level (0.92 mg/dL) and estimated glomerular filtration rate (GFR; 63.59 mL/min per 1.73 m^2^) were within the normal ranges. No proteinuria was identified. Chest computed tomography (CT) and abdominal CT were performed to exclude hidden malignancy. Chest CT revealed a subpleural nodule measuring approximately 1.5 cm in the right lower lobe. Abdominal CT revealed multiple small ill-defined low-attenuated lesions in the spleen nodules (Figure 1). Her thrombophilia screening test result was negative. The Padua prediction score was 2, in the low risk.Her echocardiography was normal and 24-hour Holter monitoring revealed normal sinus rhythm. Nerve conduction tests revealed sensorimotor polyneuropathy. The myeloperoxidase (MPO)-ANCA test result was positive (5.10 index). Percutaneous needle biopsy of the pulmonary nodule was performed first; however, the pathological results showed only focal acute inflammatory cell infiltration, with no evidence of malignancy. The second biopsy of the sural nerve demonstrated perivascular inflammatory cell infiltration and necrotizing vasculitis. These findings were consistent with vasculitic peripheral neuropathy. By considering the results of the MPO-ANCA test and the pathological profile of the sural nerve, the patient was diagnosed as having MPA-associated neuropathy with a splenic infarction and pulmonary inflammatory nodule. She received steroid pulse therapy (methylprednisolone, 125 mg/day for 3 days) on day 20 of hospitalization, followed by cyclophosphamide (500 mg/m^2^ every 4 weeks for six doses) with steroid maintenance therapy. She showed improvement after the steroid therapy. Her ESR level decreased to 49 mm/h and the CRP level decreased to 0.5 mg/dL. The peripheral numbness subsided slowly. After 1 month, the patient’s pancreatic enzyme levels (amylase, 192 U/L (23–85 U/L) and lipase 346 U/L (0–160 U/L)) were elevated without abdominal pain. Abdominal CT was performed to evaluate for pancreatitis and revealed improvement of the splenic lesion such as the infarction. The patient had an uneventful course for 5 months after the treatment.This study was approved by the Institutional Review Board (IRB) of Kyung Hee Medical Center, and the requirement for informed consent was waived (IRB 2020-12-083).

## 3. Discussion

MPA is one of the main subgroups of AAV. AAV is a group of systemic vasculitis characterized by inflammation of small- to medium-sized vessels. AAV can be divided into three entities, namely MPA, GPA, and eosinophilic GPA, according to histological findings and clinical manifestations [3,4]. Its histology is characterized by necrotizing small vessel vasculitis with little or no immune deposits [5]. In MPA, renal and pulmonary manifestations are frequent, but various other symptoms also occur, such as, fever, arthralgia, and peripheral numbness [1]. ANCA appears to be closely related to AAV, and the anti-MPO specificity of ANCAs is observed in almost 80% of patients with MPA [6]. Life-threatening manifestations are treated with glucocorticoids and cyclophosphamide. Rituximab has been shown to be effective as a remission-inducing therapy. Azathioprine, methotrexate or mycophenolate mofetil may be recommended for maintenance therapy [6].

Splenic involvement is rarely reported in AAV and tends to be undervalued. This is because most patients are asymptomatic. Gercik et al. investigated the frequency of splenic lesions in patients with AAV. In their study, splenic pathologies were observed in 19 patients (28%), of whom 7 (37%) had splenic infarction. All the patients with splenic infarction were diagnosed as having GPA [2]. No splenic infarction by MPA has been reported until recently.

I report a case of MPA with an asymptomatic splenic involvement. High-dose steroid and cyclophosphamide therapies successfully controlled the splenic infarction. Moreover, the patient had no symptoms, so infection and other causes could be ruled out. However, the lack of awareness of the possibility of splenic involvement of MPA could delay the diagnosis of vasculitis until after infection or malignancy has been ruled out.

## 4. Conclusions

Here, I describe the case of MPA presenting with splenic infarction. If practitioners overlook the splenic involvement of AAV, the diagnosis and treatment of vasculitis may be delayed until infection or other causes have been identified. Therefore, this should be kept in mind when evaluating patients with the possibility of splenic involvement in AAV, and even MPA.

## Figures and Tables

**Figure 1 medicina-57-00157-f001:**
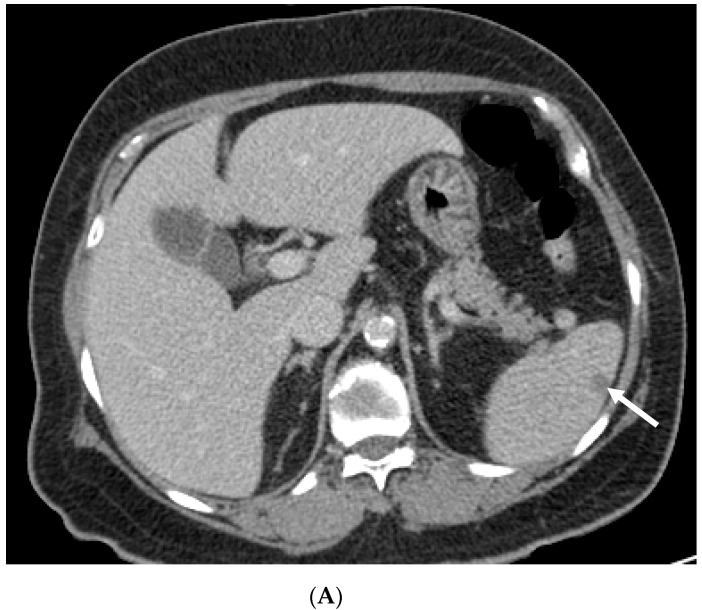
Abdominal computed tomography (CT) image. (**A**) Axial CT image on day 1, illustrating wedge-shaped splenic infarcts (white arrow). (**B**) Axial CT image 2 months later, showing an improved splenic infarction.

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
