# Peer review of "Microscopic Polyangiitis Presenting with Splenic Infarction: A Case Report"

_medicina, 2021, doi:10.3390/medicina57020157_

Round 1
Reviewer 1 Report
It is a well documented MPA case with not unusual splenic infarction in an inflammatory disease of elderly.
Classical risk factors for thrombosis should be discussed in the first place: Padua Prediction Score (or other alike scale), arteriosclerosis, possibility of portal hypertension/hepatic thrombosis, hypercoagulability or a prothrombotic state.
Please rethink the title because “Vasculitic Neuropathy” is a common manifestation of MPA.
Abstract should be more concise: it contains information on classical manifestations of MPO, but lefts a lot of potential clinical questions unanswered.
Definition is not of high quality standard because when saying that MPA “predominantly affects small- to medium-sized vessels” it equals the frequency of small and medium vessel involvement that is not correct. Please refer to 2012 Revised International Chapel Hill Consensus Conference Nomenclature of Vasculitides.
Reference ranges of lab results must be supplied.methodology must be described once it is stated that: “The patient’s motor power was normal.”.
No arrow is indicated in the figure. If splenic infarction is being described why do the authors demonstrate renal infarction in the figure?
Why do the authors believe that “High-dose steroid and cyclophosphamide therapies successfully controlled the splenic infarction”? and what does it mean “to control” the infarction?
No information on patient consent is supplied.
Minor language improvement should be performed e.g. use of plural in: “This disease can affect the kidney, nervous system, skin, and lung.”
Author Response
Response to Reviewer Comments
Thanks to carefully review the manuscript of the author.
Reviewer 1.
It is a well documented MPA case with not unusual splenic infarction in an inflammatory disease of elderly.
- Classical risk factors for thrombosis should be discussed in the first place: Padua Prediction Score (or other alike scale), arteriosclerosis, possibility of portal hypertension/hepatic thrombosis, hypercoagulability or a prothrombotic state.
- As you mentioned, I added the Padua prediction score
Page 2, Line 49
The Padua prediction score was 2, in the low risk.
- Please rethink the title because “Vasculitic Neuropathy” is a common manifestation of MPA.
- As you recommended, I retitled this article as below:
Microscopic Polyangiitis Presenting with Splenic Infarction: A Case Report
- Abstract should be more concise: it contains information on classical manifestations of MPO, but lefts a lot of potential clinical questions unanswered.
- As you mentioned, I revised the abstract more concise, and amended clinical significance of this case
Abstract: Microscopic polyangiitis (MPA) is an anti-neutrophil cytoplasmic antibody (ANCA)-associated vasculitis (AAV). The splenic involvement in AAV is known to be rare, and that in MPA has not been reported to date. A 74-year-old woman was admitted owing to left arm numbness and weakness. The patient was diagnosed as MPA with vasculitic neuropathy. Her abdominal computed tomography (CT) revealed splenic infarction incidentally. The splenic infarction had been resolved at follow-up CT after treatment. If splenic involvement of MPA was not considered, treatment may have been delayed in order to differentiate other diseases. Herein, we report the first case of splenic involvement of MPA.
- Definition is not of high quality standard because when saying that MPA “predominantly affects small- to medium-sized vessels” it equals the frequency of small and medium vessel involvement that is not correct. Please refer to 2012 Revised International Chapel Hill Consensus Conference Nomenclature of Vasculitides.
- As you commented, I changed the sentence as below
Page 1, Line 21
that predominantly affects small-sized vessels
- Reference ranges of lab results must be supplied. methodology must be described once it is stated that: “The patient’s motor power was normal.”.
- As you recommended, I added refence ranges of lab and methodology.
Page1, Line 35
the erythrocyte sedimentation rate (ESR) was 74 mm/h (normal range, < 20 mm/h)and C-reactive protein (CRP) level was 6.61 mg/dL (< 0.5 mg/dL)
The patient’s motor power of four limbs were normal (grade 5).
Page 2, Line 62
The peripheral numbness subsided slowly. After 1 month, the patient’s pancreatic enzyme levels (amylase, 192 U/L (23 – 85 U/L) and lipase 346 U/L (0 – 160 U/L))
- No arrow is indicated in the figure. If splenic infarction is being described why do the authors demonstrate renal infarction in the figure?
- As you mentioned, I added arrow on figure, and changed the renal infaction to splenic infarction
Page 3, Line 67, 68
Figure 1. Abdominal computed tomography (CT) image. (A) Axial CT image on day 1, illustrating wedge-shaped splenic infarcts (white arrow). (B) Axial CT image 2 month later, showing an improved splenic infarction.
- Why do the authors believe that “High-dose steroid and cyclophosphamide therapies successfully controlled the splenic infarction”? and what does it mean “to control” the infarction?
- It was confirmed that the lesion of splenic infarction disappeared on FU CT.
- No information on patient consent is supplied.
- Minor language improvement should be performed e.g. use of plural in: “This disease can affect the kidney, nervous system, skin, and lung.”
- As you commented, I corrected some words.
Page 1, line 22
the kidney and lung
Reviewer 2 Report
It is an interesting case describing splenic involvement as asymptomatic renal infarction with radiological expression in patient with a systemic ANCA-MPO vasculitis with predominantly neurological clinical involvement.
I have some comments:
- In the introduction (page 1) you directly talk about MPA. In line 25 you say that "However, numerous atypical manifestations may also occur, among which is splenic involvement".., but two lines below you say that "In MPA, splenic involvement has not been reported". This seems a contradiction. Text can be corrected like this, with little change: "In AAV numerous atypical manifestations may also occur, among which, though rarely reported, is splenic involvement, particularly in patients with granulomatous polyangiitis (GPA) [2]..."
- In the introduction, line 27, the statement "In MPA, splenic involvement has not been reported. Herein, we report for the first time, a case of splenic involvement of MPA"... should be revised. Although splenic involvement with clinical or radiological expression has been described mainly in patients with granulomatosis with polyangiitis, the involvement of this organ in systemic ANCA MPO vasculitis has been reported in cases with necropsy study. [As an example: Nakamura C et al. Myeloperoxidase antineutrophil cytoplasmic autoantibody-associated glomerulonephritis in a very elderly patient with generalized vasculitis at autopsy. Hiroshima J Med Sci. 1997 Sep;46(3):99-104.]
- I think that the presence of a recent cerebrovascular ischemic event together with the finding of lesions suggestive of splenic infarction makes it necessary to rule out an embolic event. To provide more consistency to the authors' assertion that the infarction is due to the vasculitis itself, it should be confirmed that the patient has a normal echocardiographic study and that she has not presented paroxysmal episodes of atrial fibrillation (for example in an ECG-Holter study).
Moreover:
- Correct the dose of the steroid bolus (page 2, line 59): 125 g/day?)
- It would be of interest to specify the cyclophosphamide regimen used (500 mg/m2, frequency?, duration?).
Author Response
Response to Reviewer Comments
Thanks to carefully review the manuscript of the author.
Reviewer 2.
- In the introduction (page 1) you directly talk about MPA. In line 25 you say that "However, numerous atypical manifestations may also occur, among which is splenic involvement".., but two lines below you say that "In MPA, splenic involvement has not been reported". This seems a contradiction. Text can be corrected like this, with little change: "In AAV numerous atypical manifestations may also occur, among which, though rarely reported, is splenic involvement, particularly in patients with granulomatous polyangiitis (GPA) [2]..."
- As you recommended, I changed the sentence
Page 1, Line 25
In AAV numerous atypical manifestations may also occur, among which, though rarely reported, is splenic involvement, particularly in patients with granulomatous polyangiitis (GPA)
- In the introduction, line 27, the statement "In MPA, splenic involvement has not been reported. Herein, we report for the first time, a case of splenic involvement of MPA"... should be revised. Although splenic involvement with clinical or radiological expression has been described mainly in patients with granulomatosis with polyangiitis, the involvement of this organ in systemic ANCA MPO vasculitis has been reported in cases with necropsy study. [As an example: Nakamura C et al. Myeloperoxidase antineutrophil cytoplasmic autoantibody-associated glomerulonephritis in a very elderly patient with generalized vasculitis at autopsy. Hiroshima J Med Sci. 1997 Sep;46(3):99-104.]
à Thanks for the great point. I changed the sentence as below,
Page 1, Line 27
In MPA, splenic involvement has been reported very rarely. Herein, we report a case of splenic involvement of MPA.
- I think that the presence of a recent cerebrovascular ischemic event together with the finding of lesions suggestive of splenic infarction makes it necessary to rule out an embolic event. To provide more consistency to the authors' assertion that the infarction is due to the vasculitis itself, it should be confirmed that the patient has a normal echocardiographic study and that she has not presented paroxysmal episodes of atrial fibrillation (for example in an ECG-Holter study).
- As you mentioned, I added the result of echocardiography and 24hrs holter monitoring.
Page 2 Line 50
Her echocardiography was normal and 24hours holter monitoring revealed normal sinus rhythm.
- Moreover: Correct the dose of the steroid bolus (page 2, line 59): 125 g/day?)
- I corrected the dose of steroid
Page 2 Line 60 methylprednisolone, 125 mg/day for 3 days
- It would be of interest to specify the cyclophosphamide regimen used (500 mg/m2, frequency?, duration?).
- As you recommended, I added the regimen of cyclophosphamide treatment
Page 2 line 61
500 mg/m2 every 4 weeks for six doses
Reviewer 3 Report
introduction part: add the upper respiratory tract (ENT) which can be affected
presentation part: the technical screening has to be detailed, and its normal range
references: there is a new paper published about de maintenance treatment:
ANCA-associated vasculitides: Recommendations of the French Vasculitis Study Group on the use of immunosuppressants and biotherapies for remission induction and maintenance, by Benjamin Terrier et al.
Author Response
Response to Reviewer Comments
Thanks to carefully review the manuscript of the author.
Reviewer 3
- introduction part: add the upper respiratory tract (ENT) which can be affected
- As you mentioned, I added about the upper respiratory tract involvement
Page 1, Line 25
upper respiratory tract involvement may be present.
- presentation part: the technical screening has to be detailed, and its normal range
à As you recommended, I added the normal range of laboratory data,
Page1, Line 36, 41, Page 2, Line 65,66
the erythrocyte sedimentation rate (ESR) was 74 mm/h (normal range, < 20 mm/h) and C-reactive protein (CRP) level was 6.61 mg/dL (< 0.5 mg/dL)
The peripheral numbness subsided slowly. After 1 month, the patient’s pancreatic enzyme levels (amylase, 192 U/L (23 – 85 U/L) and lipase 346 U/L (0 – 160 U/L))
- references: there is a new paper published about de maintenance treatment:
- As you recommended, I changed the references about treatment to a new paper and added the maintenance treatment
Page 3, Line 86
Azathioprine, methotrexate, or mycophenolate mofetil may be recommended for maintenance therapy
Page 4, References
- Terrier B, Charles P, Aumaitre O, Belot A, Bonnotte B, Crabol Y et al. ANCA-associated vasculitides: Recommendations of the French Vasculitis Study Group on the use of immunosuppressants and biotherapies for remission induction and maintenance. Presse Med2020,49,104031.
Round 2
Reviewer 1 Report
thank you for the improvements. I have no further remarks.